# SPEAK TO A PROTEIN: AN INTERACTIVE MULTIMODAL CO-SCIENTIST FOR PROTEIN ANALYSIS

## ABSTRACT

Building a working mental model of a protein typically requires weeks of reading, cross-referencing crystal and predicted structures, and inspecting ligand complexes, an effort that is slow, unevenly accessible, and often requires specialized computational skills. We introduce *Speak to a Protein*, a new capability that turns protein analysis into an interactive, multimodal dialogue with an expert co-scientist. The AI system retrieves and synthesizes relevant literature, structures, and ligand data; grounds answers in a live 3D scene; and can highlight, annotate, manipulate and see the visualization. It also generates and runs code when needed, explaining results in both text and graphics. We demonstrate these capabilities on relevant proteins, posing questions about binding pockets, conformational changes, or structure-activity relationships to test ideas in real-time. *Speak to a Protein* reduces the time from question to evidence, lowers the barrier to advanced structural analysis, and enables hypothesis generation by tightly coupling language, code, and 3D structures. *Speak to a Protein* is freely accessible at https://www.anonymousplayground.com.

**Keywords**  Agentic co-scientist, scientific discovery, molecular visualization, deep research, drug discovery, retrieval-augmented generation

## 1 INTRODUCTION

Proteins are the molecular machinery of life, and understanding their structure and function is fundamental to modern biology and medicine. For a researcher in drug discovery or molecular biology, developing an intuitive, "working mental model" of a target protein, its active sites, its conformational dynamics, and its network of interactions is a critical first step. However, this process is slow and arduous, often requiring a combination of deep domain knowledge and specialized computational skills.

A researcher investigating a protein kinase to understand how a new series of inhibitors might bind must embark on a fragmented and technically demanding workflow. This involves sifting through the PubMed literature (Sayers et al., 2020), fetching and comparing multiple structures from the Protein Data Bank (PDB) (Burley et al., 2018), querying UniProt (The UniProt Consortium, 2022) for functional annotations and disease-associated variants, and extracting structure-activity relationship (SAR) data from databases like ChEMBL (Mendez et al., 2018). Each step requires navigating different interfaces and data formats. Furthermore, deeper analysis, such as superimposing structures, identifying key interactions, or plotting bioactivity data, requires proficiency with specialized software or scripting languages, creating a significant barrier to entry for many bench scientists. This high friction for asking and answering questions restrains curiosity and slows the pace of discovery.

To address these challenges, we introduce *Speak to a Protein*, a new capability designed to transform protein analysis into an interactive dialogue with an AI co-scientist that collaborates with the user in real-time. Using recent advances in large language models (LLMs) and their evolution into multimodal scientific agents (Caffagni et al., 2024; Zhang et al., 2024; Wang et al., 2024b), our system can comprehend complex, natural language queries about a protein of interest. It autonomously retrieves, integrates, and synthesizes information from a comprehensive suite of biological data sources, including literature, structural repositories, and biochemical databases.

The core innovation of *Speak to a Protein* is its ability to ground its responses across multiple, synchronized modalities. When a user asks a question, the AI does not simply return text. It interacts with a live 3D structural viewer to highlight residues, measure distances, or annotate binding pockets. It can generate and execute Python code in a sandboxed environment to perform calculations, filter tabular data, or generate plots on the fly. Furthermore, the AI co-scientist sees, understands and controls the visualization, offering a natural interaction with the user. This tight coupling of natural language, 3D visualization, and code execution creates a seamless and intuitive environment for scientific exploration.

This paper makes the following contributions: We present the design and architecture of an AI system that integrates language, code execution, and 3D visualization for interactive protein analysis. We show how

Figure 1: **Overview of the system architecture.** The system consists of a frontend with a protein viewer and chat interface, which includes a virtual file system and Python sandbox for automated code execution and viewer manipulation. The LLM Agent is the main orchestrator that interacts with the user and calls a set of custom tools through Model Context Protocol (MCP). These tools include a literature search for a given protein and text query, retrieving specialized data through APIs such as UniProt, PDBe and ChEMBL, as well as executing Python code in a sandbox environment with a dedicated virtual file system.

this multimodal approach drastically lowers the barrier to complex structural and biochemical data analysis. Through case studies on relevant proteins, we show that this system enables a more fluid and powerful form of hypothesis generation, accelerating the cycle from question to evidence.

## 2 RELATED WORK

The ambition to create an AI capable of scientific discovery is a long-standing goal, articulated in visions such as Hiroaki Kitano's proposal for an "AI Scientist": a system that could autonomously formulate hypotheses, design experiments, and achieve Nobel-class discoveries (Kitano, 2021). While such a fully autonomous system (Boiko et al., 2023; Zou et al., 2025) remains a grand challenge, recent progress in large language models (LLMs) has enabled the development of a more immediate and collaborative paradigm: the "AI co-scientist" or "advanced intelligence" in Kitano's words.

Early systems explored conversational interfaces for structural inspection and Q&A over proteins. Guo et al. (2023) demonstrated *ProteinChat*, which couples LLM prompting with protein 3D structures to answer user questions about residues and pockets. Contemporary efforts such as Wang et al. (2024a) and Xiao et al. (2024) investigate protein-aware prompting and multimodal conditioning for function/property reasoning. Most recently, Wang et al. (2025) proposes *Prot2Chat*, an LLM that fuses protein sequence, structure, and text via an early-fusion adapter, directly targeting protein Q&A.

Beyond text-only chat, domain copilots increasingly drive molecular viewers and modeling tools. Sun et al. (2024) introduce *ChatMol Copilot*, an LLM agent that coordinates cheminformatics and modeling tools (e.g., docking, conformer generation) in response to natural-language requests. In parallel, Ille et al. (2024) systematically evaluates GPT-4's ability to perform rudimentary structural modeling and protein–ligand interaction analysis, highlighting both promise and limitations. Our work is aligned with this line but centers on tightly coupling language, code execution, and a live 3D scene for grounded, manipulable answers.

Agent frameworks augment LLMs with tool use, retrieval, and planning. *ChemCrow* (Bran et al., 2024) shows that equipping GPT-4 with chemistry tools enables multi-step synthesis planning and materials tasks. More recently, CLADD (Lee et al., 2025) proposes a retrieval-augmented multi-agent system specialized for drug discovery tasks. *Speak to a Protein* adopts the agentic paradigm for structural biology: it retrieves literature/structures, executes analyses (e.g., pocket mapping, SAR tables), and grounds responses in synchronized 3D visualizations.

Compared to prior work, our contribution is an end-to-end, *interactive* co-scientist for proteins that (i) unifies literature/structure/ligand retrieval, (ii) reasons with tabular and 3D modalities, (iii) executes code for on-the-fly analyses, and (iv) directly annotates/manipulates the 3D scene in response to dialogue (Table 1). This tightly coupled language–code–3D loop reduces the time from question to evidence relative to agent-only or text-only systems.

## 3 SYSTEM OVERVIEW: *Speak to a Protein*

### 3.1 ARCHITECTURE

The system architecture of *Speak to a Protein* (Figure 1) is organized around two main components: a frontend for user interaction and visualization, and a back-end for language understanding, tool coordination, and data retrieval. This is effectively a visual channel of communication between the AI co-scientist and the human scientist. The front-end provides the primary user interface, incorporating both a conversational

Table 1: Comparison of Speak to a Protein with representative LLM-based systems developed for drug discovery, as well as with general-purpose AI models. Speak to a Protein uniquely combines access to multiple biological databases, scientific literature extraction, on-the-fly code execution, real-time 3D molecular visualization, multimodal reasoning grounded in structural data, and persistent knowledge management. These capabilities are not collectively available in other existing platforms.

| System | API Access | Lit. Extraction | Code Sandbox | 3D Vis. Control | Multimodal Reasoning | Persistent Memory |
|---|---|---|---|---|---|---|
| Speak to a protein | Yes | Yes | Yes | Yes | Yes (Live 3D scene) | Yes |
| General-purpose AI | Not natively | Partial (cannot retrieve) | Yes | No | No | No |
| ProteinChat | No | No | No | No | Yes (Structure Description) | No |
| Prot2Chat | No | No | No | No | Yes (3D/Sequence Fusion) | No |
| ChatMol Copilot | No | No | No | Yes | Yes (Analyzes structure visualization) | No |
| ChemCrow | Yes | Yes | Yes | No | No | Yes |
| CLADD | Yes | Yes | No | No | No | No |

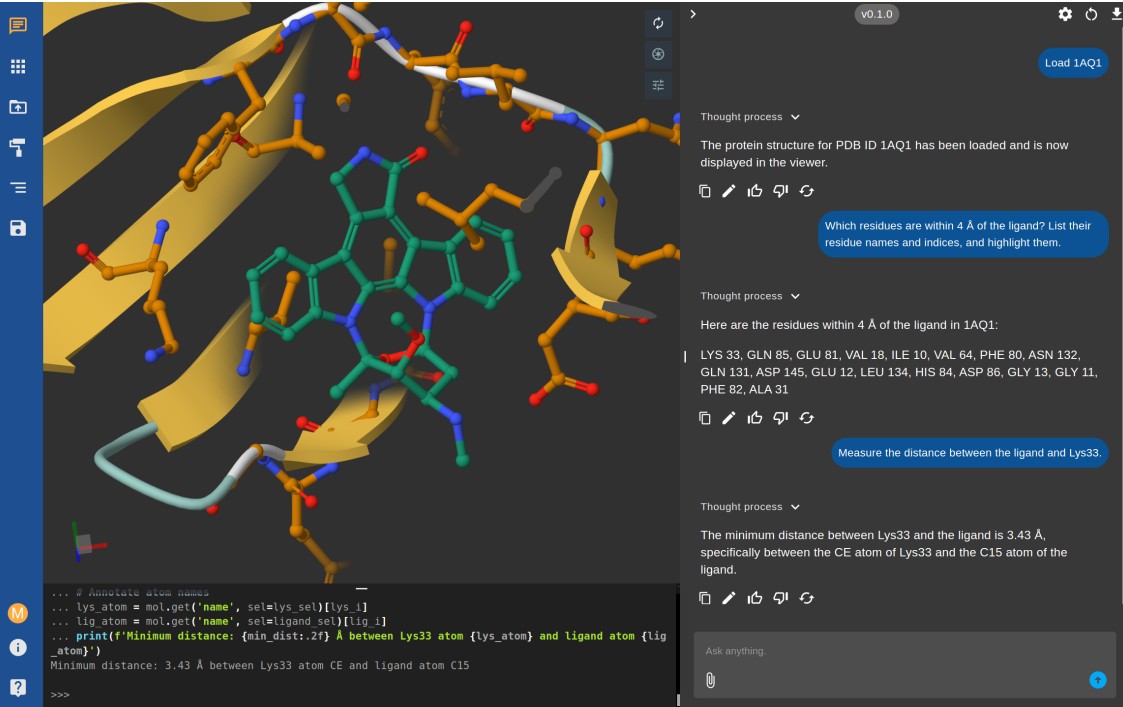

Figure 2: **Interactive analysis of the CDK2 structure (PDB: 1AQ1) using the *Speak to a Protein* multimodal assistant.** The user enters natural language queries in the AI chat panel (right), and the system responds with both textual answers and real-time updates to the 3D visualization (left). Python code executed in the integrated sandbox is shown in the console (bottom), providing full transparency and reproducibility of the underlying analyses.

chat panel and 3D molecular visualization capabilities (Figure 2 and Section 3.6). At its core is a Python sandbox powered by Pyodide, enabling the execution of Python code directly in the browser to manipulate structures and control the viewer. The sandbox includes a virtual file system, where structural files and related data are stored for visualization. Users interact through the chat panel, entering natural language

requests. The system interprets responses from the AI agent and parses them to display textual information in the chat. It also detects when a specialized action is needed and invokes the corresponding viewer tools:

- **Virtual file system tool:** Loads required data files from the backend and stores them for visualization.

- **Python tool:** Executes Python code, as generated by the model, to carry out custom analyses or visual manipulations.

The outputs from these tools are sent to the backend, which determines whether additional actions are needed or if results should be presented in the user interface. The backend processes natural language queries and orchestrates all available tools through a central AI agent, running either as a local LLM or via an external API. Upon receiving a user request, the agent plans a sequence of actions, including complex reasoning, modifying the viewer state, or invoking some of its domain-specific tools:

- **Literature Search:** Retrieves relevant scientific articles and extracts protein-related information from PubMed Central.

- **UniProt Search:** Finds protein entries and annotations, including sequence, function, and cross-references, from the UniProt database.

- **ChEMBL Search:** Retrieves bioactivity and assay data for small molecules and proteins from the ChEMBL database.

- **PDB Search:** Locates experimental 3D structures and related metadata in the Protein Data Bank.

- **MoleculeKit Search:** Enables semantic search through the source code of MoleculeKit (Doerr et al., 2016), a library for structure and formats manipulation.

- **A python sandbox**. Enables server-side computations using advanced libraries.

The system's multimodal tooling layer is implemented as *Model Context Protocol (MCP)* (Anthropic, 2024) servers that the model can invoke and compose during an interaction. Each tool provides a structured interface to a core knowledge source, with outputs designed for integration into language reasoning and context-conditioned retrieval-augmented generation (RAG). Concretely, we implement three primary tools: (i) a literature retrieval component that performs sequence- and structure-grounded searches and builds a protein-conditioned RAG corpus from PubMed Central; (ii) a UniProt interface that supports both accession discovery and access to detailed entry information, including sequence data, cross-references, functional annotations, and literature links; and (iii) a ChEMBL interface for harvesting assay activities and surfacing structure–activity relationship (SAR) information. All tools rely on programmatic access to public biological databases and return normalized, machine-readable results, enabling the model to consistently connect protein identity with structural, functional, and biochemical evidence. This modular organization allows complex scientific queries to be decomposed into well-defined tool calls, whose implementation details we describe in Sections 3.2–3.4.

## 3.2 LITERATURE SEARCH AND COMPREHENSION

The literature tool constructs a protein-specific corpus that can be searched with a text query. While the literature discovery relies on *curated references from UniProt*, the tool can be invoked with either a UniProt accession, PDB ID or FASTA sequence of the protein in question. If the user provides a UniProt accession, the references linked to this entry are augmented by expanding the selection to all linked PDB structures. If the input is a PDB code, the system first queries UniProt for all entries cross-referenced to that structure, and then collects their reference lists. If the input is a raw FASTA sequence, the system searches the Protein Data Bank to identify matching structures, resolves them to UniProt entries, and again retrieves their references. This pathway ensures that the system always incorporates the expert-curated literature that UniProt associates with a protein, regardless of the initial identifier type.

The result of the literature discovery step yields a diverse set of article identifiers, including PubMed IDs, DOIs, and PubMed Central IDs. To unify them, we normalize all entries to *PubMed Central IDs (PMCIDs)* using a public conversion service. Importantly, only the subset of publications that are openly available on PubMed Central can be downloaded and processed further; articles behind paywalls remain indexed only by their identifiers. For each accessible PMCID, we retrieve the article in XML format, which is efficient to fetch and preserves section and paragraph boundaries. The text is cleaned and segmented into coherent passages that are embedded into a vector space for retrieval-augmented generation (RAG) using LlamaIndex (Liu, 2022).

All passages for a given protein are combined into a *protein-conditioned retrieval index*, together with metadata such as PMCID, DOI, and the set of matched PDB and UniProt identifiers (Table 2). The index is cached on disk along with a list of associated protein identifiers (UniProt accessions, PDB IDs, FASTA sequences), so future queries for the same target can reuse the corpus without repeated downloads. At query time, the system formulates a descriptive retrieval prompt, retrieves the top-$k$ relevant passages, and returns them along with their citations. The retrieved text is then provided to the language model

as grounded context, either to directly answer a user's question (e.g., "Which mutations in CDK2 affect inhibitor binding?") or to supply background for further analysis and additional tool calls (e.g., filtering ChEMBL assays for compounds tested in the cited studies, or highlighting reported residues in the 3D structural viewer).

Despite involving multiple external databases and full-text retrieval, the entire pipeline runs in real time. Literature discovery, download, and RAG construction typically complete within one to two minutes for a new protein, and subsequent queries on the same target are handled within seconds due to caching of the prebuilt index.

### 3.3 UniProt: Accession Discovery and Rich Entry Information

The UniProt MCP server provides structured access to the UniProt knowledgebase, enabling both the discovery of correct accessions and the retrieval of detailed entry information. It is organized into two complementary tools. The first is a text search utility that resolves canonical entries from colloquial protein names. This call returns a concise shortlist with entry type, primary accession, protein name, organism, annotation score, and keywords, allowing the model to select the most relevant entry—for example, the reviewed entry of the correct organism—without inflating context.

Once an accession has been identified, the data lookup tool retrieves the corresponding UniProt record or resolves from a PDB identifier when structures are the starting point. The response contains identifiers and provenance suitable for citation, descriptive and gene fields, the full amino acid sequence, span-form features encoding domains, active and binding sites, post-translational modifications, and variants, compressed literature references with bibliographic fields and sequence "focus" positions, and a set of cross-references that link the UniProt entry to other databases (Table 3). These include structural repositories such as the Protein Data Bank (PDB) and AlphaFoldDB, pharmacological resources such as DrugBank and DrugCentral, and functional annotation databases such as Gene Ontology (GO). This representation is compact but expressive, preserving direct links to external resources while making it straightforward to highlight residues, align sequences, or connect functional information across databases.

In practice, the model first issues a name query (when necessary), selects appropriate entries, and then performs a data lookup to obtain a comprehensive record. The sequence can, for example, be forwarded to the literature tool for protein-conditioned retrieval; feature spans can be used to create residue highlights and distance measurements in the 3D viewer; and cross-references provide pivots to structures, pathways, or pharmacology resources. The separation of discovery (name to accession) and enrichment (accession or PDB to full entry) keeps tool contracts simple, enabling deterministic composition with other MCP servers.

### 3.4 ChEMBL: Bioactivity and SAR Tables

The ChEMBL MCP server is dedicated to retrieving and organizing information about assays recorded in the ChEMBL database. It focuses specifically on assay-level measurements of how molecules interact with a given protein target. When invoked with a target identifier and an assay type, the tool downloads all matching entries from the ChEMBL API to assemble a complete activity table. To reduce latency and ensure reproducibility, the results are cached locally; repeated queries for the same target reuse this cache until it expires.

Because raw ChEMBL activities are highly heterogeneous, mixing different units, redundant identifiers, and free-text fields, the MCP normalizes the dataset into a streamlined representation that highlights the essential biochemical information (Table 4). Rarely useful or inconsistent fields are removed, while values necessary for reasoning about structure–activity relationships, such as standard measurements, molecule identifiers, and publication context, are retained. All entries that pass this filtering are written to a tabular file in CSV format. This file is stored on disk and its path is returned alongside a compact summary object.

The stored CSV file plays an important role in the overall system. It is directly accessible in the sandboxed coding environment where the model can execute Python, enabling downstream analysis using libraries such as pandas. For example, the model can reload the file, apply additional filters, calculate statistics, or search for specific compounds that meet criteria relevant to the user's query. This design separates the heavy data retrieval and normalization step from the flexible, interactive analysis that happens later in dialogue with the scientist.

For functional and binding assays, which are the most commonly used in drug discovery, the tool also highlights the most potent entries with standardized units. This provides a quick surface view of the strongest bioactivity signals, while leaving the full dataset available for deeper inspection. Other assay families, such as ADME or toxicity, are treated similarly but are generally provided only as complete CSV tables, reflecting their diversity in format and measurement.

By structuring ChEMBL assay data into reproducible on-disk artifacts and linking them to a consistent summary interface, the MCP server makes assay information straightforward to query, analyze, and connect to the other knowledge sources in the system. It allows the model to bridge from raw assay measurements

to literature and structural contexts, supporting seamless reasoning across biochemical, structural, and sequence evidence.

## 3.5 STRUCTURAL REPOSITORIES: PDB AND PREDICTED MODELS

The PDB MCP tool provides structured access to the Protein Data Bank (PDBe) API (Burley et al., 2018). It can be invoked with one or more PDB identifiers to retrieve detailed entry information. This includes metadata such as the experimental method, resolution, and publication details, as well as molecular information like the list of co-crystallized small molecules (ligands). The tool automatically filters out common solvents and ions to return only ligands relevant for analysis, providing their chemical identifiers (SMILES, InChIKey) and cross-references to databases like ChEMBL (Table 5). This capability is crucial for large-scale structural analyses, such as identifying all ligand-bound structures for a given protein target, a common starting point in drug discovery projects.

## 3.6 MULTIMODAL GROUNDING AND INTERFACES WITH SCIENTISTS

Understanding proteins requires the navigation of multiple types of information, such as three-dimensional structures, experimental assay tables, and textual annotations. A central goal of *Speak to a Protein* is to connect these diverse modalities through a single conversational interface, ensuring that system responses are not only generated in natural language but are also grounded in concrete evidence such as 3D visualizations, data tables, and executable code. To achieve this, we provide a set of interactive interfaces that extend dialogue into complementary domains: a structural viewer for molecular inspection, a tabular analysis environment for filtering and plotting assay data, and mechanisms for synchronizing actions across views. Together, these interfaces enable users to fluidly transition between asking questions, running analyses, and visually verifying hypotheses. Significantly, we are building these capabilities on top of a web application that has already attracted more than 18,000 registered users over the years, even in the absence of these AI features. These new capabilities enable medicinal chemists with no-code experience to use the tools like an expert computational chemist. We thus anticipate that it will be used by a considerable number of scientists, as demonstrated by the platform usage metrics in Appendix A.3.

In *Speak to a Protein*, these functionalities are built using an entirely client-side sandbox for dynamic visualization and manipulation of molecular structures (Torrens-Fontanals et al., 2024). The sandbox builds on a browser-based molecular visualization toolkit that combines the high-performance mol* visualization engine (Sehnal et al., 2021), capable of rendering large biomolecular structures and molecular dynamics trajectories directly in the browser, with a WebAssembly-enabled Python runtime (Pyodide). This allows the use of powerful Python libraries such as MoleculeKit (Doerr et al., 2016) in the client environment, enhancing the viewer's capabilities to load and manipulate a wide range of common structural file formats, from PDB and CIF to molecular dynamics trajectory files such as XTC and TRR.

A key feature is the ability to control this viewer through natural language commands. Requests such as "highlight the ATP-binding site" are translated into tool calls that execute Python code within the viewer, producing visual changes in real time (Fig. 2). Users can request a broad set of actions, such as:

- **Loading structures**: Users can instruct the system to load structures either from public sources or by uploading custom files.

- **Controlling the visualization**: The AI can create, modify, or remove molecular representations on demand. Structures or any of their subsets can be rendered in diverse styles, such as cartoon, ball-and-stick, spacefill, or surface, and colored according to different properties, such as chain, residue type, secondary structure, or user-defined colors. The selection logic uses the expressive VMD selection language, enabling complex queries like "show only the protein backbone," "highlight tyrosine residues in chain A," or "display all residues within 5 Å of the ligand."

- **Focusing the viewer**: The camera can be centered or zoomed onto regions of interest, such as active sites, mutated residues, or selected domains.

- **Performing measurements**: Users can request measurements of distances, angles, or dihedrals between atoms or residues.

- **Manipulating structures**: The system can modify loaded structures, for example, by filtering out water molecules or other unwanted components, splitting chains, or extracting subsets for closer inspection.

- **Structural alignment**: Multiple structures can be superimposed based on selected atoms (e.g., C$\alpha$ atoms), allowing direct comparison of conformational states or homologous proteins.

## 4 EXPERIMENTS

We present a set of illustrative case studies to show the capabilities and versatility of *Speak to a Protein*, as well as a quantitative benchmark to validate its reliability on verifiable tasks. Firstly, we show, using

the dopamine D3 receptor (D3R) (Chien et al., 2010b) as a test case, the execution of a set of possible questions that showcase the capabilities of the platform. These examples highlight how the system enables users to ask scientific questions, integrate data from multiple sources, and rapidly generate insights through multi-modal interactive analysis. Secondly, we ask a set of interactive questions on cyclin-dependent kinase 2 (CDK2) (Malumbres & Barbacid, 2009). Finally, we ask the AI to produce a summary report in LaTeX of all the information gathered. By indexing all information, the system creates a knowledge base. Other users can then interrogate it, knowing what is information model of the protein.

## 4.1 QUANTITATIVE EVALUATION: BIO-STRUCTURAL BENCHMARK

We established a benchmark of 23 tasks designed to test capabilities essential for computational chemistry. The dataset spans four categories: (i) **Structural Geometry** (e.g., calculating RMSD, detecting steric clashes, measuring residue distances); (ii) **Sequence Retrieval** (e.g., extracting specific chain sequences); (iii) **Database Cross-Referencing** (e.g., retrieving ChEMBL potencies, identifying co-crystal ligands); and (iv) **Literature Verification** (e.g., finding specific citations). We compared *Speak to a Protein* against the standard **ChatGPT 5.1 Web (Thinking)**. This serves as a strong baseline as it is equipped with reasoning capabilities, web browsing, and a Python sandbox. However, it lacks the specialized biochemical tools, a persistent virtual file system for cross-referencing API outputs, and the 3D visualization engine integrated into our system. Tasks were scored strictly as correct or incorrect based on ground truth data derived from programmatic calculation and database entries. As shown in Figure 3, *Speak to a Protein*

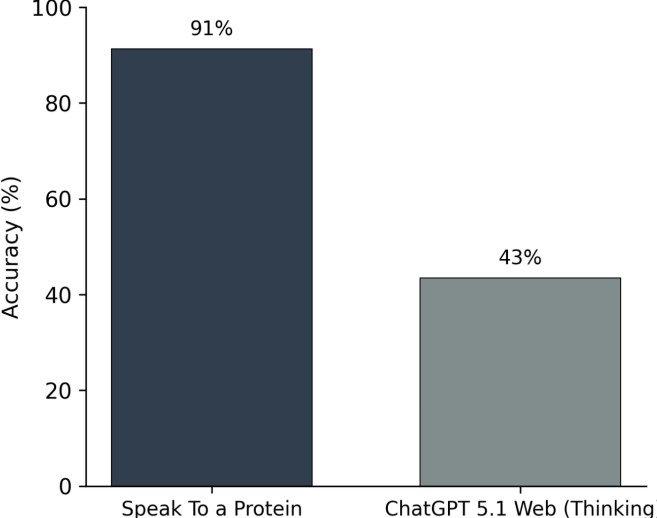

Figure 3: **Benchmark Performance.** Accuracy comparison between *Speak to a Protein* and the ChatGPT 5.1 Web (Thinking) on 23 curated bio-structural tasks.

achieved an accuracy of **91% (21/23)**, significantly outperforming the baseline at **43% (10/23)**. ChatGPT 5.1 frequently failed on tasks requiring precise structural geometry or specific bioactivity data. While it represents a powerful general-purpose resource for scientists, it lacks the specialized infrastructure that computational chemists require. Its reliance on web search often resulted in retrieval errors, and without a specialized environment to manipulate and visualize molecular structures, it could not accurately derive geometric properties. In contrast, our system succeeded by executing analysis scripts and querying domain tools, ensuring that answers were derived directly from raw data and grounded in the interactive 3D viewer. The full list of benchmark questions is provided in Appendix A.4.

## 4.2 SPEAKING ABOUT THE DOPAMINE D3 RECEPTOR (D3R)

We use *Speak to a Protein* to address a series of research questions related to D3R, a G protein-coupled receptor of significant pharmacological interest. In the video trace https://youtu.be/H6ag4JJAM0w, we show a possible user interaction with our system centered on D3R. The scientist begins by asking about the available structures for this receptor, loading the listed structure 3PBL. Next, the user instructs the system to filter for chain A, and changes the visual representation to focus on the binding pocket. Finally, it requests a list of known inhibitors associated with D3R. All this information is collected and made available to the AI so that knowledge is gathered and contextualized.

Focusing on the last query, we illustrate the workflow of the system (Figure 4). Upon the user's request, the AI used several tools in sequence to produce the necessary data. Using these tools, it identified the correct UniProt entry for D3R and used it to query ChEMBL for all relevant bioactivity data. The retrieved assay results were compiled and automatically stored in a CSV file, which was then loaded directly into the

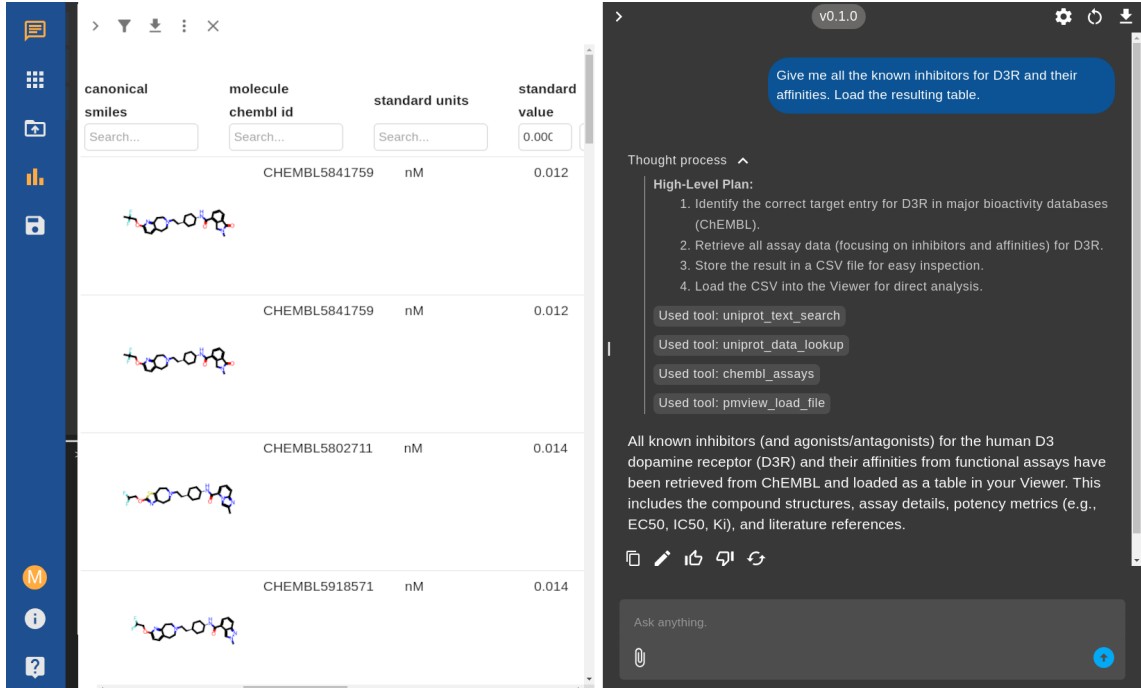

Figure 4: **Retrieval and exploration of potent D3R inhibitors.** Example user query and system workflow for fetching all known D3R inhibitors and their affinities using *Speak to a Protein*. The chat panel shows the AI's reasoning, including the tools used and the final response. On the left, the interface displays an interactive table of D3R inhibitors, including chemical structures, ChEMBL IDs, assay details, and more, enabling direct exploration and further analysis.

viewer for exploration and analysis. The resulting table provides an overview of all known D3R inhibitors, along with chemical structures, assay details, potency metrics (such as EC50, $IC50$, $K_i$), and references to the corresponding literature. Notable examples include inhibitors with subnanomolar to low nanomolar potencies such as CHEMBL5841759 ($K_i$: 0.012 nM) and CHEMBL5802711 ($K_i$: 0.014 nM). The results also listed several potent reference agonists for context.

In the video trace https://youtu.be/nER3vC90ylQ, we show how to investigate the differences between the D3 and D2 receptors. The literature search capability can help rapidly surface and synthesize expert knowledge from primary sources to address detailed structural questions. Upon receiving the prompt *"Based on the literature, compare the binding pockets of D3R and D2R and summarize the main structural features that could be exploited for ligand selectivity."*, the system first uses *UniProt Search* to identify the canonical UniProt IDs for the specified proteins, ensuring precise target selection. Then, using *Literature Search*, an embedding-based literature search is performed, retrieving relevant articles and review information from PubMed Central that address the requested topic.

In this example, the system processed a set of 12 literature passages for D2R and 10 for D3R, selecting at least four distinct, peer-reviewed structural biology studies and related supporting works (Wang et al., 2018; Chien et al., 2010a; Yin et al., 2020; Fan et al., 2020). Based on this, the generated answer highlighted that while the orthosteric binding pocket is highly conserved, selectivity can be achieved by targeting differences in the architecture and flexibility of the 'extended binding pocket', as well as in the conformation of extracellular loops. It also described the significance of distinct residues (such as Trp100 and residues in EL1/EL2, positions 1.39 and 7.35) that shape ligand binding modes and selectivity opportunities. The resulting analysis revealed that while D3R and D2R share a conserved binding pocket core, D2R possesses additional flexible and hydrophobic residues that create a deeper, more accommodating pocket. These structural differences, clearly visualized and annotated in the viewer, help explain how each receptor achieves ligand selectivity, providing actionable insights for targeted drug design.

### 4.3 SPEAKING ABOUT THE CYCLIN-DEPENDENT KINASE 2 (CDK2)

We present a complete drug discovery workflow using cyclin-dependent kinase 2 (CDK2), a validated cancer target with extensive structural and bioactivity data (Malumbres & Barbacid, 2009). This example showcases how *Speak to a Protein* can streamline the entire process from initial target evaluation to actionable insights. Our analysis begins with a systematic exploration of available structural data, progresses through mining and filtering of bioactivity data, and culminates in an integrated structural-activity analysis.

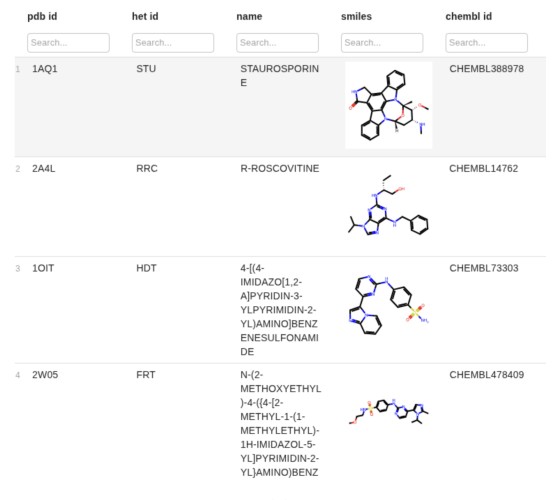

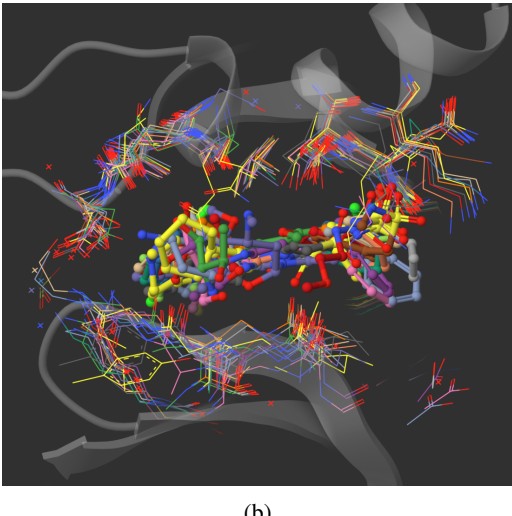

(a)                                                        (b)

Figure 5: CDK2 structure–activity analysis. (a) Data integration and bioactivity analysis derived from ChEMBL datasets, demonstrating seamless integration of structural and bioactivity data through natural language queries. (b) Structural alignment of the top 20 most potent CDK2-ligand complexes with focused visualization of ATP-binding pockets.

The workflow highlights the system's ability to seamlessly transition between different data modalities and analysis types, all through natural language interactions.

**Structure-activity.** The complete conversational trace for this analysis is provided in the Appendix A.2. First, we ask the AI system to retrieve all available CDK2 structures from the Protein Data Bank. The system first used the UniProt tool to identify the canonical human CDK2 entry (P24941) and retrieve all 462 associated PDB structures. It then systematically queried each structure using the PDB information tool, which parsed each entry to extract co-crystallized ligands. After filtering out common solvents and ions, this process identified 479 unique ligand-structure pairs containing small molecules relevant for drug discovery. The system then automatically determined bioactivity coverage, revealing that 132 out of 258 ChEMBL-annotated ligands had experimental activity measurements available. Through systematic data cleaning and deduplication focused on $IC_{50}$ values, we generated a refined dataset of approximately 100 unique CDK2-ligand complexes, ranked by potency. The top 20 most potent complexes ($IC_{50}$ values ranging from sub-nanomolar to 15 nM) were loaded into the 3D viewer and structurally aligned (Figure 5). For detailed binding site analysis, the system identified and visualized only the ATP-binding pocket residues (within 6 Å of the co-crystallized ligands), excluding solvents and common crystallization agents.

**Automated Report Generation.** Furthermore, the AI extracted the binding pocket sequences and stored them in FASTA format. Pairwise sequence alignment revealed high conservation across the ATP-binding site, with only minor variations at peripheral positions. The AI system then conducted a comprehensive literature search to contextualize these findings with existing CDK2 research, automatically generating a markdown summary of relevant studies discussing binding site features and structure-activity relationships. The following report was automatically generated by the AI system, synthesizing all gathered data into actionable insights suitable for distribution to medicinal chemistry teams:

### CDK2 Structure-Activity & Binding Pocket Analysis
### Key Findings

*Structural Data Scope:* 462 unique CDK2 PDB structures identified and curated. 479 ligand/structure pairs contain co-crystallized non-solvent small molecules suitable for drug discovery.

*Bioactivity Integration:* 258 unique CDK2 co-crystallized ligands mapped to ChEMBL IDs. 132 ligands ( 51%) have direct annotated bioactivity data for CDK2 in public databases. After deduplication, 100 unique, potent (lowest $IC_{50}$) ligand–structure matches represent the focused SAR set. Top 20 complexes span sub-nanomolar to low-nanomolar $IC_{50}$ values.

*Binding Pocket & Sequence Conservation:* The ATP-binding pocket is highly conserved across CDK2 structures, with the Gly-rich loop and DFG motif strictly maintained. Only minor sequence variations (G→V, Q/N, D/K) were observed at the pocket periphery. Pairwise sequence alignment scores are uniformly high, supporting a rigid, canonical scaffold for ligand engagement. All potent inhibitor binding modes overlay at this conserved

pocket, with activity cliffs mostly driven by ligand features rather than pocket sequence variation.

*Structural Alignment & 3D Analysis:* 14 of the top 20 potent CDK2–inhibitor complex structures were successfully aligned and visualized. Superposition reveals near-identity of main pocket conformation but highlights loop and surface flexibility for ligand-induced fit.

**Actionable Insights**

CDK2 displays a classic, highly druggable ATP-binding pocket with little sequence-derived risk of resistance. SAR optimization should focus on maximizing hydrophobic and hinge contacts, exploring diversity at the pocket periphery, and leveraging observed loop conformational plasticity for next-gen analogs.

**Data Files Generated**

| Dataset Description | File |
|---|---|
| PDB–ligand associations | druglike_ligands.csv |
| Annotated activity data | ligands_annotated.csv |
| $IC_{50}$-focused dataset | ligands_IC50_only.csv |
| Deduplicated potency table | ligands_dedup.csv |
| Pocket sequences | pocket_sequences.fasta |
| Sequence alignments | pocket_alignments.txt |
| Literature synthesis | literature_summary.md |

**Conclusion:** CDK2 remains a top-tier drug discovery target with a structurally robust, deeply conserved and well-characterized ATP site optimal for inhibitor design—validated by a wealth of crystal structures and directly observed SAR correlation.

The system's ability to generate publication-ready reports and comprehensive literature summaries further enhances its utility in collaborative drug discovery environments, where rapid communication of complex structural and biochemical insights is essential for decision-making.

## 5 CONCLUSION AND LIMITATIONS

This study demonstrates how *Speak to a Protein* compresses traditional workflows involving several hours of manual data gathering, analysis, and synthesis into an interactive session taking less than an hour. The system's ability to generate publication-ready reports further enhances its utility in collaborative drug discovery environments where rapid understanding and communication of complex structural and biochemical insights are essential for decision-making.

We found the following limitations in the current version of *Speak to a Protein*. The open web application accesses only public information, such as literature, structure-activity relationships, and structural data. This limits the data access and the understanding and downstream calculations. We plan to easily extend it with additional tools to access internal or proprietary datasets, further broadening the scope of information that can be queried, integrated. The 3D viewer can experience performance issues when rendering a large number of complex structures simultaneously. We also observe occasional difficulties in seamlessly connecting outputs between tools, stemming from different data representations across the system's components, e.g., residue indices across literature and structural databases. Furthermore, long tool outputs can strain the model's context window. Future work will focus on offloading large data payloads to files and equipping the model with a broader set of tools for file management, allowing for more robust and complex analytical workflows.

*Speak to a Protein* is freely accessible at https://www.anonymousplayground.com, which guarantees the reproducibility of the results shown.

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

# A APPENDIX

## A.1 TABLES

Table 2: Fields returned by the Protein Literature MCP. Global fields describe the query context; results contain one or more retrieved citations with text passages.

| Field | Description / Example Value |
|---|---|
| fasta_sequences[] | One or more FASTA sequences associated with the protein input. |
| pdb_ids[] | PDB identifiers discovered for the input (via cross-references or sequence search). *Example:* ["1H1R", "3DDQ", "7RWE"] |
| uniprot_ids[] | UniProt accessions associated with the input. *Example:* ["P24941"] |
| results[].pmcid | PubMed Central ID for a retrieved article. *Example:* "PMC5291709" |
| results[].doi | Digital Object Identifier for the article. *Example:* "10.7554/eLife.20818" |
| results[].content | Retrieved text passage from the article body (cleaned paragraphs). *Example:* "ATP-binding sites involve Walker-A and -B motifs and a trigger residue that regulates hydrolysis." |
| error | Error message if something went wrong; empty/absent otherwise. *Example:* "Failed to load existing RAG database: . . . " |

Table 3: Fields returned by the UniProt MCP. Two panels are shown: the upper panel lists fields from the **text search** tool (multiple candidate entries for a text query such as "CDK2"); the lower panel lists fields from the **data lookup** tool (detailed entries for a UniProt accession or PDB identifier).

| Field (per entry) | Description / Example Value |
|---|---|
| **Panel A — Text search tool** | |
| entries[].entry_type | UniProt record type (reviewed/unreviewed). |
| entries[].uniprot_accession | Primary accession. *Example:* "P24941" (human CDK2). |
| entries[].knowledgebase_id | Human-readable ID. *Example:* ["CDK2_HUMAN", "CDK2_MOUSE"]. |
| entries[].name | Recommended, short and alternative protein names. *Example:* "Cyclin-dependent kinase 2", "p33", "Cell division protein kinase 2". |
| entries[].organism | Common and scientific names. *Example:* "Human — Homo sapiens". |
| entries[].annotation_score | Curation score string. *Example:* "5.0 out of 5". |
| entries[].keywords[] | Keyword strings (category: name). *Example:* ["Ligand: ATP-binding", "Molecular function: Kinase"] |
| **Panel B — Data lookup tool** | |
| primaryAccession | Canonical accession. *Example:* "P24941". |
| uniProtkbId | Human-readable ID. *Example:* "CDK2_HUMAN". |
| entryType | Record type (reviewed/unreviewed). |
| organism | Common and scientific names. *Example:* "Human — Homo sapiens". |
| proteinDescription | Recommended full name and alternatives. *Example:* "Cyclin-dependent kinase 2", alt: ["p33 protein kinase"]. |
| genes[] | Gene symbols and synonyms. *Example:* ["CDK2", "CDKN2"]. |
| comments[] | Curated comment blocks (FUNCTION, SUBUNIT, PTM, ...). *Example:* FUNCTION describes CDK2 roles at G1/S. |
| features[] | Sequence features ("type \| start:end"). *Example:* "Active site — 127:127". |
| references[] | Bibliography (title, first author, date, journal, refs, focus). |
| crossref[] | External cross-refs as "Database:ID". *Example:* ["PDB:1H1Q", "DrugBank:DB07755"]. |
| keywords[] | Structured keywords with category and name. *Example:* ["Molecular function: Kinase"]. |
| sequence | Full amino acid sequence. *Example:* "MENF. . .HRL". |
| annotationScore | Numerical annotation score. *Example:* 5.0. |
| proteinExistence | Evidence level. *Example:* "1: Evidence at protein level". |
| secondaryAccessions[] | Secondary accessions, if any. |
| entryAudit | Versioning and dates. *Example:* first public 1992-03-01; last update 2025-06-18. |
| extraAttributes | Derived counts (comment/feature types) and UniParc ID. |
| error | Error message if something went wrong; empty otherwise. |

Table 4: Fields returned by the ChEMBL MCP. Global fields describe the assay query context and where results are stored; entries contain per-assay measurements with assay metadata, compound identifiers, and activity values.

| Field | Description / Example Value |
|---|---|
| assay_type | Type of assay retrieved from ChEMBL. *Example:* "functional". |
| data_csv_path | File path on disk to the cached CSV table of results, available to the LLM in a sandboxed coding environment. |
| num_entries | Total number of entries returned by the query. *Example:* 927. |
| top_n | Number of top entries actually included in the response. *Example:* 20. |
| entries[].assay_chembl_id | ChEMBL identifier of the assay. *Example:* "CHEMBL663764". |
| entries[].assay_description | Description of the assay protocol. *Example:* "Percentage A2780 cells in sub-G1 after 24 hr at 330 nM (IC50)". |
| entries[].assay_type | Short assay type code. *Example:* "F". |
| entries[].bao_label | BAO (BioAssay Ontology) label describing the assay format. *Example:* "cell-based format". |
| entries[].molecule_chembl_id | ChEMBL identifier of the tested compound. *Example:* "CHEMBL428690". |
| entries[].molecule_pref_name | Preferred molecule name, if available. *Example:* "ALVO-CIDIB". |
| entries[].canonical_smiles | Canonical SMILES string of the compound. |
| entries[].document_chembl_id | Identifier of the source publication in ChEMBL. *Example:* "CHEMBL1135763". |
| entries[].document_journal | Journal of publication (if known). *Example:* "J Med Chem". |
| entries[].document_year | Year of publication (if known). *Example:* 2002. |
| entries[].standard_relation | Relation operator for the reported value. *Example:* "=". |
| entries[].standard_type | Type of activity measured. *Example:* "Ki", "Cell cycle". |
| entries[].standard_units | Units of measurement. *Example:* "nM". |
| entries[].standard_value | Reported numeric value. *Example:* "1.0", "100.0". |

Table 5: Fields returned by the PDB MCP. The tool can fetch information for a single PDB ID or a list of them. For a single ID, it returns a dictionary; for multiple, a list of dictionaries.

| Field | Description / Example Value |
| --- | --- |
| pdb_id | PDB identifier. *Example:* "1AQ1" |
| title | Title of the PDB entry. |
| experimental_method | Experimental technique used. *Example:* "X-RAY DIFFRAC-TION" |
| resolution_angstrom | Resolution of the structure in angstroms. *Example:* 2.1 |
| publication_year | Year of the primary publication. |
| authors | List of authors of the publication. |
| journal | Abbreviated journal name. *Example:* "Cell" |
| doi | Digital Object Identifier for the publication. |
| deposit_date | Date the structure was deposited. |
| release_date | Date the structure was released. |
| small_molecules[] | List of dictionaries, one for each co-crystallized small molecule ligand. |
| small_molecules[].het_id | Heterogen ID of the ligand. *Example:* "STU" |
| small_molecules[].name | Name of the ligand. *Example:* "STAUROSPORINE" |
| small_molecules[].smiles | SMILES string of the ligand. |
| small_molecules[].inchi_key | InChIKey of the ligand. |
| small_molecules[].chembl_id | ChEMBL ID of the ligand, if available. *Example:* "CHEMBL388673" |

## A.2 FULL CONVERSATIONAL TRACE FOR CDK2 ANALYSIS

This section provides the complete, unedited conversational trace between the user and the *Speak to a Protein* system for the CDK2 case study.

**User:** I'm starting a new research project focused on CDK2 (Cyclin-dependent kinase 2). Can you retrieve all available PDB structures associated with CDK2?

**Assistant:** Great! Your goal is to get all available PDB structures for CDK2 (Cyclin-dependent kinase 2). Here's the high-level plan:
**Checklist for Retrieving CDK2 PDB Structures**

1. Identify the correct UniProt accession code for human CDK2.

2. Cross-reference this accession with the PDB database to get all associated structures.

3. Provide you with the list of PDB IDs (optionally, titles or other metadata).

Let's start by finding the UniProt accession for human CDK2.

```
tool_execution: uniprot_text_search
tool_execution: uniprot_data_lookup
tool_execution: head
tool_execution: head
```

**Summary of Actions:**

- Identified the human CDK2 UniProt accession as P24941.

- Retrieved the cross-reference table of all PDB entries linked to this protein.

- Confirmed there are over 460 PDB structures associated with human CDK2.

- The first several PDB IDs include: 1AQ1, 1B38, 1B39, 1BUH, 1CKP, 1DI8, 1DM2, 1E1V, 1E1X, 1E9H, 1F5Q, 1FIN, 1FQ1, 1FVT, 1FVV, 1G5S, 1GIH, 1GII, 1GIJ.

**What's Next:** I can provide you with the full CSV containing all CDK2 PDB IDs for your research records, or perform further filtering (e.g., by resolution, ligand-bound states, etc.) if you have specific needs.
Would you like the complete list as a CSV download, or do you wish to explore/filter these structures further?

**User:** Great! Now I need to identify which of these CDK2 structures contain bound small molecules relevant for drug discovery. For each PDB structure with a co-crystallized small molecule ligand, please extract the ligand information, including the SMILES string and ChEMBL ID.

**Assistant:** Your aim is to filter the 460 CDK2 structures for those with co-crystallized small-molecule ligands relevant for drug discovery, and to gather key ligand information (SMILES, ChEMBL ID).

```
tool_execution: run_python
tool_execution: pdb_info
tool_execution: run_python
tool_execution: pdb_info
tool_execution: head
tool_execution: run_python
```

All CDK2 PDB structures have been screened for co-crystallized small-molecule ligands, explicitly excluding common solvents and ions. Here's what was accomplished:

- A total of 462 CDK2 structures were checked.
- 479 unique ligand/structure pairs were identified that contain relevant small molecules suitable for drug discovery workflows (excluding water, ions, and common crystallization agents).
- For each entry, key information—PDB ID, ligand three-letter code, ligand name, SMILES string, and ChEMBL ID (when available)—has been extracted.

You can now download the structured CSV file with these results for your research: `cdk2_structures_with_druglike_ligands.csv`.
Do you want an overview table here, a full downloadable link, or further sub-filtering (e.g., by ligand/chemotype, ChEMBL presence, etc.)?

**User:** How many of our co-crystallized ligands now have annotated bioactivity data?

**Assistant:** Your question aims to determine how many of the CDK2 co-crystallized ligands identified (from the previous CSV, each with ChEMBL IDs when available) have associated, annotated bioactivity data.
Here's the high-level plan:
**Checklist to Address Ligand Bioactivity Data**

1. Extract the list of unique ChEMBL IDs among the co-crystallized ligands.
2. Query ChEMBL for bioactivity records for these ChEMBL IDs.
3. Count and report how many ligands have at least one bioactivity annotation.

Let's start by extracting the unique, non-null ChEMBL ligand IDs from the results file.

```
tool_execution: run_python
tool_execution: chembl_assays
tool_execution: run_python
tool_execution: head
```

```
tool_execution: run_python
```

**Result:** Out of 258 unique CDK2 co-crystallized small-molecule ligands with a ChEMBL ID, 132 have annotated bioactivity data in ChEMBL.

This means just over half of the ChEMBL-annotated ligands seen crystallographically with CDK2 have corresponding experimental activity measurements available—enabling robust structure-activity relationship (SAR) and virtual screening workflows.

If you'd like, I can generate a CSV file listing all ligands with/without bioactivity data, or prioritize by type/affinity ranges—just let me know your next analytic need!

**User:** Excellent! Now let's load this dataframe into the viewer so I can explore and analyze the data interactively.

**Assistant:** Perfect approach! Here's how we'll proceed:
**Checklist for Data Loading & Interactive Analysis**

1. Prepare the relevant CSV with CDK2 PDB-ligand-activity data for the viewer.

2. Load the CSV into the Viewer's VFS (frontend).

3. Enable you to filter, sort, and explore SAR and ligand info interactively from the Viewer's data table interface.

I will ensure the full co-crystal ligand dataset is loaded (with PDB ID, ligand name, SMILES, ChEMBL ID, etc.), so you can immediately begin your exploration. The entire CDK2 structure–ligand dataset (including all PDB IDs, ligand names, SMILES, and ChEMBL IDs for co-crystallized druglike compounds) is now loaded into the Viewer. You can interactively filter, sort, and analyze the data directly in the viewer's table interface to support your structure–activity relationship (SAR) and drug discovery workflows.

If you need to overlay additional columns, merge in the bioactivity values, or automate further analyses, just let me know!

```
tool_execution: pmview_load_file
```

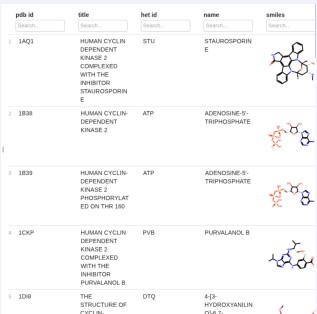

**User:** Could you annotate the dataframe with the activity data for each compound?

**Assistant:** Your goal is now to enrich the current dataframe—listing all CDK2 co-crystal ligands—with their experimental activity data, where available. Here's the plan:
**Checklist for Activity Data Annotation**

1. For each ligand in the dataframe (by ChEMBL ID), retrieve associated CDK2 bioactivity values from ChEMBL.

2. Merge the experimental activity data (e.g., $IC_{50}$, $K_i$, units, assay type, etc.) into the original structure–ligand CSV.

3. Reload the updated, annotated CSV into the viewer for interactive inspection.

**Approach:**

- I will extract activity records from the CDK2 ChEMBL assay CSV for all ChEMBL-mapped ligands in your PDB file.

- Merge these activity annotations back into the master ligand/PDB dataframe.

- Load the resulting enriched file for interactive SAR analysis.

Let's proceed with merging the activity data and preparing the annotated file. Your CDK2 co-crystal structure–ligand dataframe is now fully annotated with experimental activity data—including standard type ($IC_{50}$, $K_i$, etc.), values, units, assay description, publication, and year—for each compound where available.

The annotated CSV is already loaded and ready for interactive sorting, filtering, and SAR analysis in the viewer.

If you want to visualize, group by, or further analyze any specific fields (e.g., color by $IC_{50}$, filter by activity type), just let me know your next analytic step!

```
tool_execution: run_python
tool_execution: pmview_load_file
```

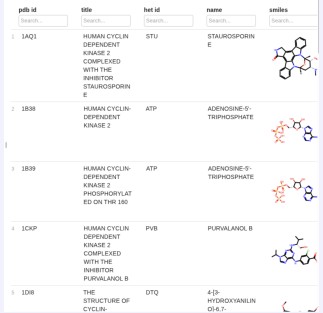

**User:** The dataset looks comprehensive but has some redundancy. Let's clean it up by removing duplicate entries and focusing specifically on $IC_{50}$ values, since these are the most commonly used and comparable activity measurements for structure-activity relationship analysis in drug discovery.

**Assistant:** Great—focusing on $IC_{50}$ values and removing redundancy will streamline your SAR analysis. Here's the plan:

**Checklist for Cleaning & Focusing the Dataset**

1. Filter the annotated dataset to retain only rows with `standard_type = 'IC50'`.

2. Remove duplicate entries (e.g., same PDB–ligand–$IC_{50}$) to keep only unique records.

3. Reload the cleaned and focused dataset into the viewer for efficient, interpretable analysis.

I'll prepare the filtered file and update the viewer. The dataset has now been filtered to include only unique entries with experimentally measured $IC_{50}$ values, the gold standard for SAR in drug discovery. Redundant rows have been removed, resulting in a concise dataset (353 unique records) now reloaded and ready for further interactive analysis in the Viewer.

If you want to further cluster by ligand, select only the best (lowest) $IC_{50}$ per compound, or need additional summary statistics or visualizations, just let me know!

```
tool_execution: run_python
tool_execution: pmview_load_file
```

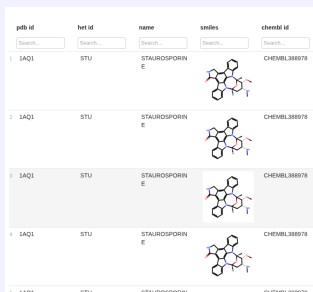

**User:** The data still has multiple entries per structure. Let's deduplicate further by keeping only unique PDB and ChEMBL IDs with the lowest $IC_{50}$ value.

**Assistant:** Excellent refinement! Here's how we'll obtain a clean, one-to-one mapping:

**Checklist for Strict Deduplication**

1. For each PDB structure (PDB ID), keep only the ligand/row with the lowest $IC_{50}$.

2. For each ChEMBL ligand, keep only the entry with the lowest $IC_{50}$ across all PDBs (to ensure only unique ligands remain).

3. Ensure the final dataframe contains only unique PDB IDs and unique ChEMBL IDs, each associated with their most potent (lowest) $IC_{50}$ measurement.

4. Reload the deduplicated dataset for your analysis.

Let's process and reload this improved, non-redundant summary. The dataset is now strictly deduplicated—containing only unique PDB IDs and unique ChEMBL ligands, each paired with its most representative (lowest) $IC_{50}$ value. This streamlines your SAR analysis by maximizing coverage and minimizing redundancy.

The resulting ~100-entry file is loaded and ready for direct exploration in the Viewer. If you'd like additional summaries, descriptor calculation, or clustering, let me know your next analysis goal!

```
tool_execution: run_python
tool_execution: pmview_load_file
```

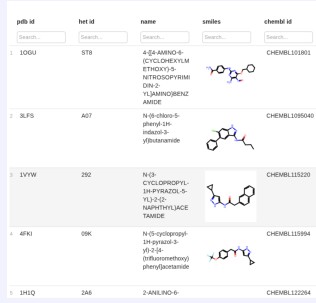

**User:** Can you now sort the table by activity?

**Assistant:** Here's the plan for your request:

**Checklist for Table Sorting by Activity**

1. Sort the deduplicated dataset by the $IC_{50}$ value in ascending order (most potent at the top).

2. Reload the sorted file into the Viewer so you can immediately interact with it in this order.

This will let you quickly identify the most potent (lowest $IC_{50}$) CDK2 co-crystalized ligands directly from the Viewer interface. Proceeding with sorting and reloading now. The table is now sorted by $IC_{50}$ (ascending)—the most potent CDK2 binders appear at the top. You can immediately explore these prioritized results in the Viewer to focus your SAR hypotheses and compound design work.

If you want to visualize specific compounds, compare chemical scaffolds, or export subsets for follow-up, just specify your next action!

```
tool_execution: run_python
tool_execution: pmview_load_file
```

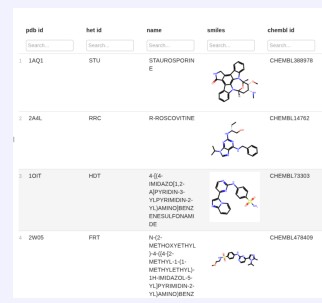

**User:** Perfect! Now I want to visualize the top 20 most potent complexes in 3D.

**Assistant:** Great—your goal is to visualize the top 20 most potent CDK2–ligand complexes (lowest $IC_{50}$) in 3D, loading their PDB structures so you can analyze binding modes and structural determinants of potency.

**Checklist for Visualization of Top Complexes**

1. Extract the top 20 PDB IDs from the sorted SAR dataframe.

2. Load each PDB structure into the viewer using the system's molecule-handling class.

3. Sequentially render all structures, allowing you to toggle, compare, and inspect binding geometries and features.

I'll retrieve the top 20 PDB IDs and then load/visualize them in the Viewer. It appears there was an error when loading some of the top 20 PDB structures—likely due to missing files, unsupported formats, or transient PDB server issues.

Here's what I'll do next to robustly address this:

- I will re-attempt loading each structure individually, catching and skipping any that fail, ensuring you see as many successful 3D visualizations as possible for SAR analysis.

I'll proceed with a revised loading script that summarizes any problematic entries and gives you a maximally complete visual set. **Visualization Complete:**

- 14 out of the top 20 most potent CDK2–ligand complexes were successfully loaded and rendered in 3D for your structural analysis.

- The following PDBs could not be loaded, possibly due to PDB server issues or unsupported formats: 2WPA, 1KE7, 5IEV, 4FKQ, 2VU3, 4FKV.

You can now examine and compare the highest-potency binding modes interactively in the viewer. If you'd like to troubleshoot the unavailable entries or focus on any specific structures, just let me know!

```
tool_execution: run_python
tool_execution: pmview_run_python
tool_execution: pmview_run_python
```

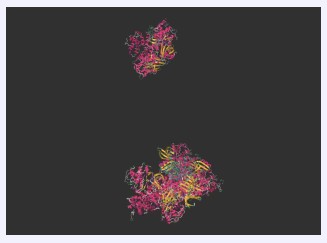

**User:** I have loaded the remaining structures manually. Now please visualize all loaded structures with structure-index representation and remove the waters.

**Assistant:** Here's the step-by-step approach for your request:

**Checklist:**

1. For each loaded structure, switch the protein representation to "structure-index" color (clarifies chain/domain/feature diversity).

2. Remove all water molecules from every PDB structure (to declutter the binding pocket visualization).

3. Ensure the ligand and relevant cofactors remain for clear analysis.

Let's proceed to update all currently loaded structures as requested. All loaded CDK2 structures have been updated as follows:

- Waters are removed from every complex.

- Proteins are visualized in cartoon representation colored by structure-index, highlighting chain and domain diversity.

- Ligands and cofactors remain, shown as ball-and-stick with element coloring.

- Views are automatically focused on the protein for each structure.

You can now clearly compare potent binding modes and structural features across your top complexes for SAR interpretation. Let me know if you'd like to further highlight specific residues, pocket regions, or overlay ligands!

`tool_execution: pmview_run_python`

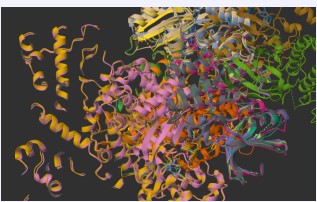

**User:** Now that all structures are loaded, please align them using structural superposition.

**Assistant:** Absolutely! Structural alignment (superposition) of all loaded CDK2 complexes will allow you to meaningfully compare the binding sites, conformational dynamics, and ligand positioning across your top SAR hits.

**Checklist for CDK2 Structural Superposition**

1. Select a reference CDK2 structure (e.g., the most potent or the first loaded structure).

2. Align all other CDK2 protein chains to the reference using structural superposition (typically on C atoms or the protein backbone).

3. Update the visualization so that all complexes share the same spatial frame, facilitating pocket and ligand comparisons.

I'll proceed by performing the superposition, using the protein backbone for robust alignment. It looks like the superposition function was not successfully imported, which suggests that superposition might either need to be called differently or is located elsewhere in the library's API.

To resolve this, I will search the library's codebase for the correct usage or import path for structural superposition/alignment and then apply the correct method to align all loaded CDK2 complexes. The library provides a built-in method for superposition:

- Use the `align` method: `mol.align(sel, refmol, refsel)`
    - `sel`: selection string for atoms to align (e.g., 'protein and backbone' or 'name CA')
    - `refmol`: Molecule to align to (reference)
    - `refsel`: atom selection for reference molecule (should match sel for structure-based alignment)

Example usage (from documentation):

`mol.align('protein', refmol=ref, refsel='protein')`

This will superpose each molecule (`mol`) onto the reference (`ref`) using the selected atoms.

I'll now perform this alignment for all currently loaded CDK2 structures using backbone atoms for maximal structural comparison. The alignment attempt failed due to differing numbers of backbone atoms between structures, which is common for PDBs with varying constructs, sequence boundaries, or missing residues.

To fix this, I'll use the library's align method with `mode='structure'`, which performs a structural alignment regardless of atom order or counts, making it robust when sequence/atom mismatches exist.

Let's proceed with structure-based alignment for the superposition. All CDK2 complexes are now structurally aligned using robust structure-based superposition. This enables direct, meaningful comparison of binding modes, conformational variability, and ligand placement across your top potent co-crystal structures.

You can now examine the aligned ensemble in the viewer for detailed SAR and structure-function insights. Let me know if you'd like to focus on active site residues, overplot specific ligands, or conduct further analyses!

```
tool_execution: pmview_run_python
tool_execution: codebase_moleculekit_search
tool_execution: pmview_run_python
tool_execution: pmview_run_python
```

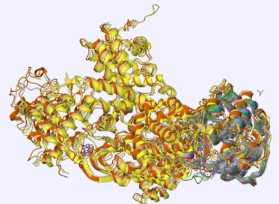

**User:** Perfect! Now, for each structure, display only the ATP-binding pocket residues (within 6Å of the ligand) and hide the rest of the protein.

**Assistant:** Here's how we'll zoom in on the ATP-binding site for detailed analysis:

1. For each aligned CDK2 structure, identify all protein residues within 6 Å of the co-crystallized ligand.

2. Update the visualization to display only these ATP-pocket residues (hide all other parts of the protein).

3. Keep the co-crystallized ligand(s) visible with ball-and-stick representation.

This will allow direct, uncluttered comparison of pocket residues and ligand interactions across all aligned complexes for SAR and binding analysis.

Let's execute these selections and update your visualization accordingly. All aligned CDK2 structures are now focused solely on the ATP-binding pocket—showing only those protein residues within 6 Å of their co-crystallized ligand, together with the ligands themselves.

This view enables you to directly compare key pocket residues and their interactions with high-potency ligands across the series, making SAR, hotspot, and mechanism-of-action assessments far more efficient.

Let me know if you'd like to further highlight conserved residues, map mutations, overlay multiple ligands, or export pocket coordinates for computational analysis!

```
tool_execution: pmview_run_python
```

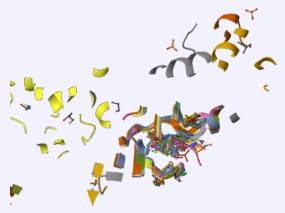

**User:** Do not consider solvents or ions like SO4, CH3, SGM, GOL, ACT, ACE as ligands for the pocket selection.

**Assistant:** Understood! For ATP-pocket residue selection, we'll strictly exclude all common crystallization agents, solvents, and ions—such as SO4, CH3, SGM, GOL, ACT, ACE, and others you listed. Only true organic/inhibitor ligands will define the selection.

**Checklist:**

1. Exclude all common/solvent/ion three-letter codes from ligand identification.

2. For each structure, update the selection so the ATP pocket is defined by protein residues within 6 Å of only the remaining true ligand(s).

3. Keep only those pocket residues and the organic ligands visible.

Let's refine the selections accordingly and update the viewer. All pocket views have now been refined: Only protein residues within 6 Å of true co-crystallized organic/inhibitor ligands are displayed, strictly excluding all common solvents and ions (e.g., SO4, CH3, SGM, GOL, ACT, ACE, EDO, waters, metals). Ligands themselves remain clearly visible.

You can now directly analyze the critical ATP-binding environments and ligand interactions across all top SAR cases—without any distracting non-relevant species.

Let me know if you'd like to highlight conserved residues, compare ligand orientations, or export pocket coordinates for downstream modeling!

```
tool_execution: pmview_run_python
```

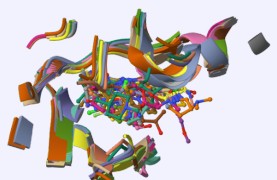

**User:** Could you extract the amino acid sequences of each of the binding pockets?

**Assistant:** To extract the amino acid sequences of the ATP-binding pockets for each structure, we need to:

**Checklist**

1. For each structure, identify all protein residues within 6 Å of the (properly filtered) ligand(s).

2. Retrieve the unique residue numbers and one-letter codes for these residues.

3. Output or save the binding pocket sequences (one per structure), which you can compare, align, or analyze further.

I will perform these steps and present the results in a tabular form for your review. You can use the library's `sequence` method, which has a selection argument (`sel`) to return the one-letter amino acid sequence for a given region, such as the ATP-binding pocket:

Example usage:

```
pocket_sequence = mol.sequence(oneletter=True, sel=pocket_sel)
# Returns a dictionary: {segid: sequence}
```

I'll apply this for each structure's identified pocket and display the resulting one-letter sequences for direct comparison. It appears there was an error when extracting the sequences—likely because, in some structures, the selected ATP pocket region is empty or not protein (e.g., no residues within 6 Å, or the segment definition fails).

To solve this, I'll adjust the script to handle cases with empty selections gracefully, so we can present the sequences for all successfully detected ATP pockets without failing on missing ones.

Here are the ATP-binding pocket amino acid sequences (one-letter code, for residues within 6 Å of the filtered ligand) from each structure:

```
| PDB ID | Pocket Sequence (one-letter code) |
|--------|:----------------------------------|
| 1aq1   | IGEGTGVAKVFEFLHQDKPQNLLADL        |
| 3ddq   | EKIGEGVKALKVFEFLHQDKQNLLADEK...   |
| 1oit   | IGEGGVVAKVFEFLHQDLKFQNLAD         |
```

```
| 2w05   | IGEGVAKVFEFLHQDLKFQNLLAD          |
| 2w17   | IGEVAKVFEFLHQDKKDQNLLAD           |
| 2vtp   | IEGTGVVALKVFEFLHQDKQNLLADL        |
| 1r78   | IGEGTVVAKVKFEFLHQDKQNLADF         |
| 3ns9   | EKIGEGTGVKALKVLFEFLHQDKQNLAD      |
| 4bgh   | IGEVAKVFEFLHQDLKFQNLLAD           |
| 2c6o   | IGEGTGVAKVFEFLHQDKFKQNLAD         |
| 1ke6   | IGGVAKVFEFLHQDKQNLAD              |
| 6gue   | IGEYVAKEVFEFLHQDKQNLLAD           |
| 6gue   | IGEGTYVAKEVFEFLHQDLKQNLLAD        |
| 1ckp   | EIVAKVFEFLHQDKQLAD                |
| 1g5s   | EIGEGTVKALKVFEFLHQDDKQNLAD        |
| 2wpa   | IGYVALKEVLFEFLHQDKKQNLADIGYV...   |
| 1ke7   | IEGTYGVAKVFEFLHQDKFKQNLADL        |
| 5iev   | EIGEVALKVKFEFLHQDKFQNLLAD         |
| 2vu3   | IVVALKVFEFLHQDKQNLLAD             |
```

These sequences correspond to the local environment of the ATP site in each structure, capturing subtle sequence or conformation differences. Let me know if you want these saved/downloadable, aligned, or mapped onto your structures for further comparative analysis!

```
tool_execution: pmview_run_python
tool_execution: codebase_moleculekit_search
tool_execution: pmview_run_python
tool_execution: pmview_run_python
```

**User:** Thanks! Could you write them into a FASTA file?

**Assistant:** The ATP-binding pocket amino acid sequences for your CDK2 structures have been written into a single FASTA file:
`cdk2_pocket_sequences.fasta`
You can now download or use this file for sequence comparison, clustering, motif analysis, or further bioinformatics workflows. Let me know if you need a multiple sequence alignment or conservation analysis!

```
tool_execution: run_python
```

**User:** Now please analyze the conservation patterns and sequence variations in these binding pocket sequences.

**Assistant:** Let's synthesize the insights from your ATP-pocket sequences and structural ensemble:
**High-Level Analysis Plan**

1. Examine sequence conservation and variation across the ATP-binding pockets.

2. Relate observed patterns to ligand binding modes and measured activities.

3. Highlight any meaningful residue changes (substitutions, insertions, or deletions) and their possible structural/functional impacts on ligand binding.

**1. Conservation Patterns**
Looking at your ATP-pocket FASTA sequences, there's strong conservation across the binding site—the core motif `IGEGTGV(A/G)KVFEFLHQ(D/K)(K/Q)QNLLAD(L/F)`—reflecting the critical, evolutionarily fixed nature of the ATP site in CDK2.
Most of the sequence changes are conservative (G→V, D→K, Q→N, etc.), clustered at the periphery, and rarely directly at residues known to contact ligands or substrate.

**2. Key Sequence Variation Observations**

- A few sites (notably G→V, D/K/Q exchanges) appear as minor variations across structures, with some longer sequences showing apparent repetition or possible unresolved loop insertions (e.g., in 3ddq or 2wpa).
- The strong conservation of the "VFEFLHQD" region highlights its central role in ATP or inhibitor recognition; this is further supported by its near invariance in all structures.

**3. Correlation with Binding Modes/Activities**

- Because most potent ligands (lowest $IC_{50}$) are co-crystallized with structures with identical or nearly identical pocket sequences, it reinforces that activity cliffs are driven more by ligand features and conformational fit than sidechain mutations in these structures.
- Where variations exist at or near the pocket (e.g., E→Q or G→V), these changes may:
    - Alter the shape or polarity/hydrophobicity of the local environment
    - Induce repositioning of main/sidechains affecting direct or water-bridged hydrogen bonds, key for some ligand classes
    - Potentially shift flexibility/"breathability" of the pocket size—favoring larger or more conformationally dynamic ligands

If a specific structure's ligand shows a marked difference in binding orientation (e.g., unique rotations, hydrogen bonding, or stacking), and that structure's sequence has a rare variant in the pocket, this is a plausible hypothesis for the divergence in SAR.

**4. Implications for Drug Design**

- The deep conservation means CDK2 actives will likely recognize the same pocket features—the real SAR differentiation will be tied to microenvironments, local water networks, and dynamic sidechains.
- Subtle variations (Q/N, D/K) could lead to favorable/unfavorable interactions for designed inhibitors, especially those with polar moieties.
- For any observed outlier in binding mode or $IC_{50}$, check if it coincides with an unusual local sequence change—it may be actionable for the next round of analog design.

**Next Steps:**

- I can generate a pairwise alignment or conservation plot for visual mapping.
- If you'd like, I can map the sequence variations back onto your 3D structures for spatial correlation with ligand pose differences.

Let me know your analytical or visualization preference for the next step!

---

**User:** Generate a pairwise alignment.

---

**Assistant:** A comprehensive pairwise alignment of all CDK2 ATP-pocket sequences has been generated. Here's what you can do next:

- You can view the full file (all alignments, scores, and residue matches):
  `cdk2_pocket_pairwise_alignments.txt`
- The alignments quantify pairwise similarity (alignment scores) and precisely map where insertions, deletions, or substitutions occur between each pocket.

**Key Observations**

- The core ATP-binding motif is highly conserved across nearly all your potent SAR structures.
- Most pairwise similarity scores are high, with only rare single-residue or small motif deviations.
- Insertions and rare longer repeats (e.g., in 3ddq, 2wpa) may reflect flexible loops or alternate segmentations, possibly influencing local conformational plasticity or ligand accommodation in outlier cases.

**Practical Next Steps**

- Download and review the full alignment to correlate specific sequence changes with structural or activity outliers.
- Highlight in the viewer any pocket residues that vary in the pairwise alignments, for structure-guided design insights.

If you'd like assistance visualizing the conservation or highlighting variable residues directly on your aligned 3D ensembles, I can guide you through that next!

`tool_execution: run_python`

---

**User:** Based on our structural and sequence analysis findings, particularly the conservation patterns we identified and the sequence variations in the ATP-binding pocket, please search the literature for CDK2 to find relevant studies that discuss these binding site features, structure-activity relationships, and any reported effects of pocket variations on inhibitor binding. This will help contextualize our findings with existing knowledge. Write this into a .md file

---

**Assistant:**

`tool_execution: query_protein_literature`

`tool_execution: write_file`

Your literature synthesis has been compiled into a markdown file: `cdk2_pocket_literature_summary.md`.

---

**User:** Based on our analysis, please generate a concise summary report for the head of chemistry.

---

**Assistant:**

`tool_execution: write_file`

A concise executive summary for the head of chemistry has been prepared. You can download the report here: `CDK2_analysis_summary.md`.

## A.3 USAGE METRICS

To demonstrate the practical utility and robustness of the system, we analyzed usage statistics from the public deployment of *Speak to a Protein* between May 2025 and November 2025. These metrics reflect organic adoption by the scientific community. As shown in Figure 6, the platform has seen consistent growth in its user base. Simultaneously, the volume of interaction has accelerated. Figure 6b displays the number of questions processed per week, which remained steady at approximately 100 queries/week during the initial months and surged to over 700 queries/week by November.

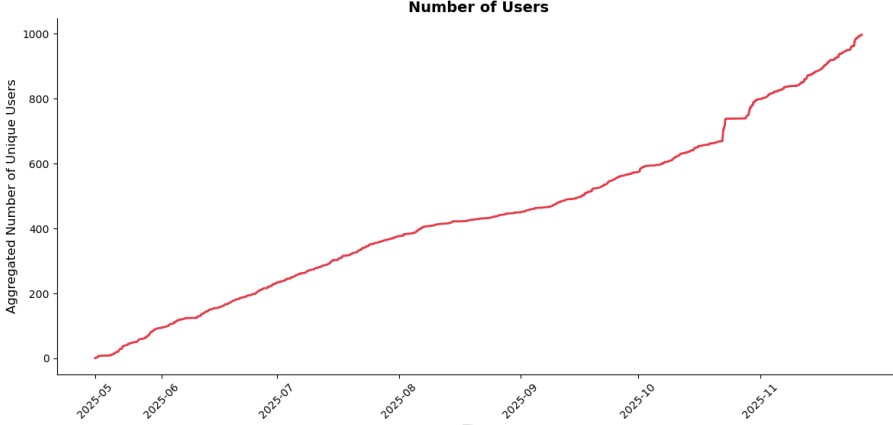

(a) Aggregated number of unique users engaging with the AI Chat (May–Nov 2025).

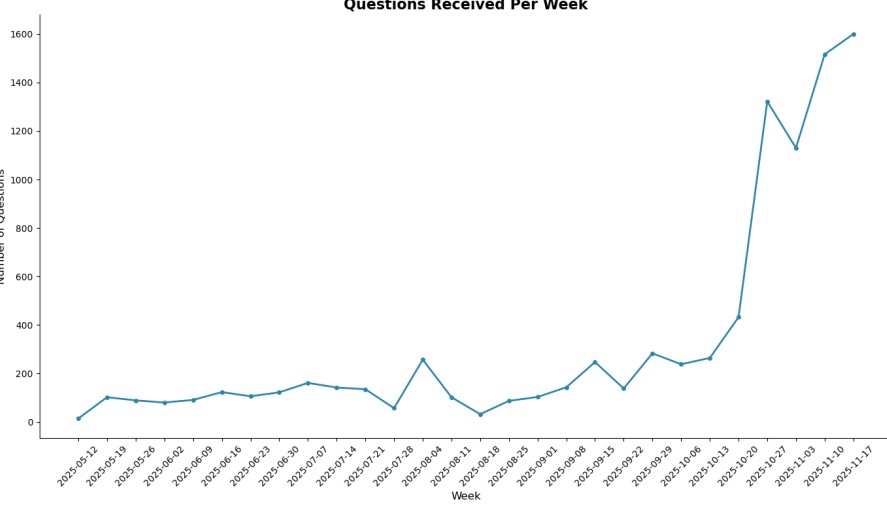

(b) Weekly volume of natural language queries received by the system.

Figure 6: **Platform Usage Statistics.** Usage metrics from the public release of *Speak to a Protein*, demonstrating steady user adoption and a significant increase in query volume over the last six months.

## A.4 Bio-Structural Benchmark Questions

We provide the full list of 23 questions used in the quantitative benchmark (Section 4.1). Results are categorized as **Correct** or **Incorrect** (including incomplete or partially wrong answers). Tasks 1–12 focus on structural queries and database retrieval, while tasks 13–23 include complex analysis, calculation, and literature synthesis.

Table 6: Benchmark Tasks 1–12. Comparison of success rates on structural geometry and identification.

| ID | Question | Speak To A Protein | ChatGPT 5.1 Web (Thinking) |
|---|---|---|---|
| 1 | For PDB 3L9H (Kinesin Spindle Protein), using the bound ligand EMQ as the reference, which protein residues have at least one atom within 6.0 Å of any EMQ atom? Return the residue list. | Correct | Incorrect |
| 2 | For PDB 1AQ1 (CDK2), using the bound inhibitor STU as the reference, which protein residues have at least one atom within 6.0 Å of STU? Return the residue list. | Correct | Incorrect |
| 3 | In PDB 1LYZ (lysozyme), what is the shortest O···O distance (Å) between any side-chain carboxylate oxygen of Glu35 and any side-chain carboxylate oxygen of Asp52? | Correct | Incorrect |
| 4 | Using HIV-1 protease structures 1HHP (apo) and 1HVR (ligand-bound), align the proteins (backbone-based). After superposition, do any heavy atoms of the 1HVR ligand come within 2.0 Å of any heavy atom of flap residues 46–56 of 1HHP (i.e., a clash)? | Correct | Correct |
| 5 | For human CDK2, which residue must be phosphorylated for full kinase activation? Answer with residue name and number. | Correct | Correct |
| 6 | What is the primary ABL1 gatekeeper mutation associated with resistance to ATP-competitive inhibitors, and what is the corresponding residue name/number in PDB 2HYY? | Correct | Correct |
| 7 | What is the full amino acid sequence of chain A in PDB 1AQ1 (CDK2)? Return it in FASTA-style (header + sequence). | Correct | Correct |
| 8 | In PDB 1AQ1 (CDK2) with ligand STU, what is the shortest heavy-atom distance (Å) between any protein atom and any ligand atom? | Correct | Incorrect |
| 9 | For human EGFR, list compounds with $IC_{50} <$ 10 nM. Provide at least 10 examples with: ChEMBL ID, name, $IC_{50}$ (nM), and SMILES. | Correct | Incorrect |
| 10 | Identify the 5 most potent human EGFR inhibitors by $IC_{50}$ (lowest, normalized to nM). Report if there is a PDB co-crystal structure of EGFR bound to that inhibitor, and list PDB IDs. | Correct | Incorrect |
| 11 | Find a PDB structure of influenza neuraminidase bound to oseltamivir (Tamiflu). What are the PDB ID and the reported resolution (Å)? | Correct | Incorrect |
| 12 | Identify a human dopamine D3 receptor (DRD3) co-crystal structure in the PDB. Report the PDB ID and the bound non-solvent small-molecule ligand. | Correct | Correct |

Table 7: Benchmark Tasks 13–23. Comparison on complex analysis, calculation, and literature synthesis.

| ID | Question | Speak To A Protein | ChatGPT 5.1 Web (Thinking) |
|---|---|---|---|
| 13 | From ChEMBL bioactivity data for human DRD3, retrieve $K_i$ measurements (nM). Provide a table of at least 20 entries with metadata (IDs, SMILES, assay/doc references). | Correct | Incorrect |
| 14 | Using CDK2 inhibitor complexes PDB 1AQ1 and 1H1Q, align 1H1Q onto 1AQ1 using protein $C\alpha$ atoms. What is the RMSD (Å) over the aligned $C\alpha$ atoms? | Correct | Incorrect |
| 15 | In PDB 1AQ1, list all non-protein hetero groups. Indicate which are solvents/ions vs. true ligand(s), and justify. | Correct | Correct |
| 16 | In PDB 3NY8, considering ligand CAU, list residues forming H-bonds, salt bridges, and hydrophobic contacts ($\leq$ 4.0 Å). Report the count for each category. | Correct | Incorrect |
| 17 | How many distinct PDB structures are available for human CDK2 in the Protein Data Bank? Provide the count and description. | Correct | Correct |
| 18 | From ChEMBL, identify the 5 most potent CDK2 inhibitors by $IC_{50}$ (normalized to nM). Report values and compute their mean (nM). | Correct | Incorrect |
| 19 | For PDB 2HYY (ABL1), identify the bound non-solvent ligand(s). List protein residues within 6.0 Å (chain, name, number). | Correct | Incorrect |
| 20 | For the ligand in PDB 1HVR, compute RDKit-style descriptors: MW, cLogP, HBD, HBA, and TPSA. | Incorrect | Correct |
| 21 | Compare the binding-pocket size in PDB 3L9H (ligand EMQ) vs CDK2 PDB 1AQ1 (ligand STU). Which is larger and what quantitative proxy was used? | Incorrect | Incorrect |
| 22 | From structural literature on CDK2, which hinge-region residues are most commonly reported to form key H-bonds with inhibitors? | Correct | Correct |
| 23 | From structural literature comparing D3R and D2R, what binding-site features are reported to enable selective ligand design? Summarize and cite. | Correct | Correct |

