# OpenReview forum: "Speak to a Protein: An Interactive Multimodal Co-Scientist for Protein Analysis"
_ICLR.cc/2026/Conference — ICLR 2026 Conference Desk Rejected Submission_

### Official Review · Reviewer_9Zca · 2025-10-27

**Soundness:** 2
**Presentation:** 2
**Contribution:** 2
**Rating:** 2
**Confidence:** 4

**Summary:**

This paper presents an interactive multimodal AI "co-scientist" system called “Speak to a Protein,” designed to simplify protein analysis through natural language dialogue. The system integrates literature retrieval, structural data, and biochemical databases (such as UniProt, PDB, and ChEMBL), and combines real-time 3D visualization with code execution to provide an intuitive analytical platform. Through case studies—such as the dopamine D3 receptor and CDK2 protein—the study demonstrates the system’s capabilities, including highlighting binding pockets, measuring distances, and generating structure–activity relationship (SAR) tables. The primary contribution of this work lies in the design of an AI co-scientist system that significantly lowers the barrier to complex structural analysis.

**Strengths:**

The “Speak to a Protein” system demonstrates originality in protein analysis by integrating natural language processing, real-time 3D visualization, and automated code execution into a unique interactive framework. Compared to existing systems, its capabilities in grounding natural language queries in real-time 3D molecular scenes and automating code execution are particularly distinctive. The system exhibits high quality, effectively coordinating diverse tools through the Model Context Protocol (MCP), and its reliability is further validated through comprehensive case studies. The paper is also highly clear in presentation: its structure logically progresses from an overview of the system, through tool design, to experimental case studies, and includes intuitive visualizations that effectively illustrate the interactive workflow.

**Weaknesses:**

- The current paper reads more like a technical demonstration (demo paper). It primarily focuses on showcasing system capabilities, such as multimodal interaction and 3D visualization, while offering limited discussion of novel underlying algorithms or model architectures.
- Although “Speak to a Protein” integrates multimodal functionalities, its core concept, language-driven protein analysis, has already been explored in prior work, such as ProteinChat and Prot2Chat. The authors should clearly articulate their specific technical advances and provide quantitative comparisons against these existing systems.
- The experimental design leans toward illustrative demonstration rather than rigorous validation. The case studies (e.g., D3R and CDK2) serve primarily as intuitive showcases of functionality, rather than systematic evaluations of the system’s performance, generalizability, or robustness.

**Questions:**

1. The paper mentions distinctions from ProteinChat and Prot2Chat but does not provide a detailed functional comparison highlighting their limitations. Could the authors offer a concrete example demonstrating in what specific aspects—such as real-time 3D grounding, code execution integration, or multimodal responsiveness—“Speak to a Protein” significantly outperforms these existing systems?

2. The current evaluation is limited to only two proteins (D3R and CDK2). Do the authors plan to expand testing to a broader range of protein types and develop a standardized benchmark to enable systematic and quantitative assessment of the system’s capabilities?

---

> ### Author Response · Authors · 2025-11-28
>
> The conference's scope includes topics such as "infrastructure, software libraries, hardware, systems, etc,” as indicated on the conference website. System papers are well within the scope of the conference. Furthermore, this “system” is open and free to use, with thousands of registered scientists, so its impact is significant. We have added usage metrics in Appendix A.3 to demonstrate this utility.
>
> Regarding **Question 1**, we have provided a detailed comparison (Table 1) of Speak to a Protein with other LLM-based systems for drug discovery and biology, including ProteinChat and Prot2Chat, as well as with general-purpose AI tools. This comparison highlights that, while prior work has explored language-driven protein analysis, **Speak to a Protein** is the first to combine real-time, AI-driven 3D molecular visualization, integration of multiple biological databases and literature sources, persistent knowledge management, and the ability to execute and visualize code dynamically. These multimodal and agentic capabilities, taken together, represent a significant advancement over existing platforms and are not collectively found elsewhere.
>
> | System            | API Access   | Lit. Extraction              | Code Sandbox | 3D Vis. Control | Multimodal Reasoning                    | Persistent Memory |
> |------------------|--------------|------------------------------|--------------|------------------|------------------------------------------|-------------------|
> | Speak to a protein | Yes          | Yes                          | Yes          | Yes              | Yes (Live 3D scene)                      | Yes               |
> | General-purpose AI | Not natively | Partial (cannot retrieve)    | Yes          | No               | No                                       | No                |
> | ProteinChat        | No           | No                           | No           | No               | Yes (Structure Description)              | No                |
> | Prot2Chat          | No           | No                           | No           | No               | Yes (3D/Sequence Fusion)                 | No                |
> | ChatMol Copilot    | No           | No                           | No           | Yes              | Yes (Analyzes structure visualization)   | No                |
> | ChemCrow           | Yes          | Yes                          | Yes          | No               | No                                       | Yes               |
> | CLADD              | Yes          | Yes                          | No           | No               | No                                       | No                |
>
> To address **Question 2**, we added a new section on benchmark-style evaluation designed to test chemistry and structure-based tasks. We compare **Speak To a Protein** to “ChatGPT5-5.1 (thinking) with Python and Websearch” as a strong general chatbot baseline. Each answer is scored as correct/incorrect based on whether it matches the required output type and is verifiably correct (e.g., residue lists computed from the structure, numerical values within reasonable tolerance where applicable, correct PDB IDs/resolution, correct ChEMBL id, etc.). On this benchmark:
>
> - **Speak to a Protein:** 91% accuracy
> - **ChatGPT-5.1:** 43% accuracy
>
> This is summarized in the new section of the manuscript.

---

### Official Review · Reviewer_ZUwP · 2025-10-29

**Soundness:** 3
**Presentation:** 3
**Contribution:** 2
**Rating:** 4
**Confidence:** 3

**Summary:**

This paper presents an LLM-based AI system, referred to as Speak to a Protein, which transforms protein analysis into a multimodal dialogue. The system comprises a frontend for user interaction and visualization, and a backend responsible for language understanding, tool coordination, and data retrieval. Experiments on D3R and CDK2 demonstrate the effectiveness of the proposed system.

**Strengths:**

1. The proposed system provides an intuitive and easy-to-use interface, making protein analysis more accessible to a broad range of users.

2. The paper is clearly written and well organized, which facilitates understanding of the system design and experimental results.

**Weaknesses:**

The main weakness of this paper lies in its limited scientific contribution. The work is primarily an engineering effort, focusing on system design rather than addressing or resolving a concrete scientific problem. Moreover, based on the presented experiments, the proposed “co-scientist” agent functions mainly as a research assistant, facilitating existing workflows rather than contributing to genuine scientific discovery. It remains unclear whether the system is capable of generating new scientific insights or findings.

**Questions:**

Please compare this work with existing LLM-based systems for scientific applications, particularly those in biology and drug discovery. What are the technical advantages or novel contributions of this paper relative to prior work?

---

> ### Author Response · Authors · 2025-11-28
>
> We would like to emphasize that the conference’s scope explicitly includes topics such as infrastructure, software libraries, hardware, and systems. We introduce a system with innovative capabilities, such as real-time, multimodal integration of literature, protein structure, and biochemical data, interactive 3D visualization, and programmatic analyses, that are not available together in other platforms. Given that the entire application is freely accessible to anyone, including commercial and academic institutions, the reader can take advantage of the novel functionalities and overall capabilities described in the paper and reuse them for their own data.
>
> To highlight the technical advantages and novel contributions of our work, we provide a detailed comparison of Speak to a Protein with representative LLM-based systems developed for drug discovery (Table 1 of the updated manuscript), as well as with general-purpose AI models. This comparison demonstrates that Speak to a Protein uniquely integrates real-time 3D molecular visualization, access to multiple biological databases and literature, persistent knowledge management, and on-the-fly code execution within a unified, interactive environment. These capabilities are not collectively available in other existing platforms.
>
> | System              | API Access  | Lit. Extraction       | Code Sandbox | 3D Vis. Control | Multimodal Reasoning            | Persistent Memory |
> |---------------------|-------------|------------------------|--------------|------------------|---------------------------------|-------------------|
> | Speak to a protein   | Yes         | Yes                    | Yes          | Yes              | Yes (Live 3D scene)             | Yes               |
> | General-purpose AI   | Not natively| Partial (cannot retrieve) | Yes       | No               | No                              | No                |
> | ProteinChat          | No          | No                     | No           | No               | Yes (Structure Description)     | No                |
> | Prot2Chat            | No          | No                     | No           | No               | Yes (3D/Sequence Fusion)        | No                |
> | ChatMol Copilot      | No          | No                     | No           | Yes              | Yes (Analyzes structure visualization) | No          |
> | ChemCrow             | Yes         | Yes                    | Yes          | No               | No                              | Yes               |
> | CLADD                | Yes         | Yes                    | No           | No               | No                              | No                |

---

### Official Review · Reviewer_JepS · 2025-10-31

**Soundness:** 3
**Presentation:** 3
**Contribution:** 2
**Rating:** 4
**Confidence:** 3

**Summary:**

This paper introduces a system that tries to probe or dialog, through a interactive interface, about a protein. The system integrates an LLM that has access to several tools and databases (Uniprot, PDB, ChEMBL, PubMed etc) with a molecular viewer. Using a model context protocol, the LLM coordinates tool calls, returns literature, and even executes python code. The Speak to the Protein system parses and understands the user's natural language queries and then plans multi-step reasoning chains which are that executed via tool calls or direct responses from the LLM. The paper presents case studies for the dopamine D3 receptor, and the CDK Kinase.

**Strengths:**

- This paper takes a leap in the enabling a close integration between natural language, code execution, and 3D visualization. The ambition of this work is quite a leap over previous works like ProteinChat, and ChatMol Copilot.
- The work provides a fully developed prototype that is accessible for people to test online.
- Appendix provides traces, which is commendable.

**Weaknesses:**

First, I should say that I don't have biochemistry or deep protein knowledge to be able to judge the case studies. In this case, my review and discussion here is more suited to the system design. I think this work should be better positioned as a systems design paper instead of a methods paper.

- Since I cannot judge the degree to which the case studies provide novel or difficult to find insight, I would've liked to see some form of quantitative measure of success of the system other than such case studies.
- Overall, this paper combines several known tools and approaches to build an impressive system. However, there is no new insight in this work. The paper could strengthen its contribution by formalizing its planning or tool-selection strategy, or by reporting insights about how LLMs behave as scientific agents (e.g., success rates, error modes).

**Questions:**

- How does the system decide which tool to call? In what order? Is there an explicit planning step or is this delegated to the LLM?
- How are conflicts handled between data sources? For instance, if ChEMBL and UniProt report different ligand mappings or residue numbering, how does the system resolve discrepancies?
- How often do failures occur?
- I would push the authors towards more quantitative evaluation via a potential human-in-the-loop validation.
- How often does the system produce incorrect, incomplete, or inconsistent results across repeated runs?

---

> ### Author Response · Authors · 2025-11-28
>
> The conference's scope includes topics such as "infrastructure, software libraries, hardware, systems, etc,” as indicated on the conference website. System papers are well within the scope of the conference. Furthermore, this “system” is open and free to use, with thousands of registered scientists, so its impact is significant. We have added usage metrics in the Appendix
>
> **How does the system decide which tool to call? In what order? Is there an explicit planning step or is this delegated to the LLM?**
> In our system, the planning and selection of tools are delegated to the LLM agent, which interprets the user’s query and generates a short-term plan, typically outlining which tools to invoke and in what order. For longer-term planning, as may be requested by the user for multi-step workflows, the LLM can store plans in its file system, allowing them to be modified before execution or revisited in future sessions.
>
> **How are conflicts handled between data sources? For instance, if ChEMBL and UniProt report different ligand mappings or residue numbering, how does the system resolve discrepancies?**
> Data source conflicts are also handled by the LLM. The system either reconciles the data based on context or, when ambiguity arises, surfaces provenance details and prompts the user to resolve or clarify as needed.
>
> **How often do failures occur? I would push the authors towards more quantitative evaluation via a potential human-in-the-loop validation. How often does the system produce incorrect, incomplete, or inconsistent results across repeated runs?**
> To address concerns regarding the correctness and success rate of our system, we have incorporated a new evaluation section (Section 4.1) into the manuscript, performing a benchmark-style evaluation designed to test chemistry and structure-based tasks.
>
> We designed a set of 23 questions (listed in Appendix 4) that are structural biology and drug discovery workflows. The questions are intentionally not hard for a bioinformatician or computational chemist (they are standard structure/database lookups and simple geometry computations), but they typically require direct access to structured data, 3D visual information and coordinate-based calculations rather than text-only reasoning. As a result, they are a useful check of whether the system can reliably execute these routine workflows through tools, code, and the molecular viewer.
>
> Our questions set spans:
> - structure-derived geometry and contact queries (distances, clashes, RMSD, residue-within-radius lists),
> - sequence retrieval from a specific PDB chain,
> - database-grounded bioactivity and structure cross-referencing (e.g., ChEMBL potency queries, “is there a co-crystal structure?”),
> - literature-backed questions requiring citations and domain-specific interpretation.
>
> We compared **Speak To a Protein** against a strong general-purpose chatbot baseline, ChatGPT5-5.1 (thinking) with Python and Websearch. Each answer was scored as correct / incorrect based on whether it matched the required output type and was verifiably correct (e.g., residue lists computed from the structure, numerical values within reasonable tolerance where applicable, correct PDB IDs/resolution, correct ChEMBL id, etc.).
>
> The results of this benchmark were as follows:
> - **Speak to a Protein:** 91% accuracy
> - **ChatGPT-5.1:** 43% accuracy
>
> In conclusion, our internal validation suggests Speak to a Protein provides a consistent and correct scientific assistant for tool-grounded tasks, with an architecture designed to minimize failures and manage the complexity of real-time, multimodal data processing. These results are summarized in the new section in the manuscript, Section 4.1.
>
> Additionally, to highlight the technical advantages and novel contributions of our work, we also provide a detailed comparison of Speak to a Protein with representative LLM-based systems developed for drug discovery, as well as with general-purpose AI models (Table 1).

---

### Official Review · Reviewer_Eo5v · 2025-10-31

**Soundness:** 3
**Presentation:** 4
**Contribution:** 3
**Rating:** 2
**Confidence:** 2

**Summary:**

The paper presents a LLM agent system connected to multiple protein-related MCP servers and visualization. It is designed as a co-scientist to provide protein analysis.

**Strengths:**

The proposed system looks well designed and created. The overall UI and the motivation behind the system is well supported by the industry as far as I know. The explanation is clear.

**Weaknesses:**

I don't see any particular weakness besides the paper is about a system, rather than a research and experiment. I think it is a great system, but it is hard to determine the value to present this as a paper in a conference. What would be the learning for the readers? It would be a different story if the product is open-sourced, but it doesn't seem to be the case.

**Questions:**

1 - L 47, 48: Citing for LLMs would be tricky, but I'm not sure I'd agree with citing GPT-4 technical report.

---

> ### Author Response · Authors · 2025-11-28
>
> The conference's scope includes topics such as "infrastructure, software libraries, hardware, systems, etc,” as indicated on the conference website. While system papers are less common, they are still within the scope of the conference. Furthermore, this “system” is open and free to use and has thousands of registered scientists, so the impact is significant. Reproducibility of the results is possible via the free web app, for which an anonymous version for reviewers has been made available, while the public version is available now. We have included in the paper appendix A.3 with usage stats of the last few months, to give a metric for utility.
>
> This is the first co-scientist that can communicate interactively, bidirectionally and graphically with the human scientist, making it a co-scientist. This is supported by a new set of tools that make it possible:
>
> 1) **A bi-directional visualization interface:** Enables the agent to manipulate the viewer’s internal state by generating executable python code, a novel methodology for ‘grounding’ LLM outputs in physical 3D space.
> 2) **Specialized MCPs:** That integrate heterogeneous biological and chemical data into structured context and tables for reasoning.
> 3) **A sandboxed execution environment in the browser and one in the backend.** This allows the agent to generate and run Python code for data analysis as well as control the visualization.
>
> We cited gpt4 because that’s what we were using but we have changed the citation to:
> Using recent advances in large language models (LLMs) and their evolution into multimodal scientific agents \citep{Caffagni2024, Zhang2024, SciDiscovery2025}, our system can comprehend complex, natural language queries about a protein of interest.
>
> We also added a new section on benchmark-style evaluation designed to test chemistry and structure-based tasks. We compare **Speak To a Protein** to “ChatGPT5-5.1 (thinking) with Python and Websearch”  as a strong general chatbot baseline. Each answer is scored as correct / incorrect based on whether it matches the required output type and is verifiably correct (e.g., residue lists computed from the structure, numerical values within reasonable tolerance where applicable, correct PDB IDs/resolution, correct ChEMBL id, etc.). On this benchmark:
>
> - **Speak to a Protein:** 91% accuracy
> - **ChatGPT-5.1:** 43% accuracy
>
> This is summarized in the new section in the manuscript.
>
> To highlight the technical advantages and novel contributions of our work, we also provide a detailed comparison (Table 1) of Speak to a Protein with representative LLM-based systems developed for drug discovery, as well as with general-purpose AI models.

---

### Note · Program_Chairs · 2026-01-17
**Submission Desk Rejected by Program Chairs**

The following references in this submission do not refer to real documents and/or have major errors in bibliographic information:

 Y. Zhang et al. From automation to autonomy: A survey on large language models in scientific discovery. arXiv preprint arXiv:2405.13259, 2024.